# *CHEK2* Pathogenic Variants in Greek Breast Cancer Patients: Evidence for Strong Associations with Estrogen Receptor Positivity, Overuse of Risk-Reducing Procedures and Population Founder Effects

**DOI:** 10.3390/cancers13092106

**Published:** 2021-04-27

**Authors:** Paraskevi Apostolou, Vasiliki Dellatola, Christos Papadimitriou, Despoina Kalfakakou, Elena Fountzilas, Eleni Faliakou, Georgios Fountzilas, Ourania Romanidou, Irene Konstantopoulou, Florentia Fostira

**Affiliations:** 1Molecular Diagnostics Laboratory, INRaSTES, National Center for Scientific Research “Demokritos”, 15341 Athens, Greece; apostolouv@hotmail.com or v.dellatola@rrp.demokritos.gr (V.D.); d.kalfakakou@rrp.demokritos.gr (D.K.); reena@rrp.demokritos.gr (I.K.); 2Oncology Unit, Second Department of Surgery, Aretaieion Hospital, National and Kapodistrian University of Athens, 11528 Athens, Greece; cpapadim@med.uoa.gr; 3Oncology Unit, General Clinic-Hospital “Euromedica”, 54645 Thessaloniki, Greece; Fountzila@oncogenome.gr; 4Second Breast Unit, Mitera Private Hospital—Breast Center, 15123 Athens, Greece; faliakou@faliakou.gr; 5Laboratory of Molecular Oncology, Hellenic Foundation for Cancer Research, Aristotle University of Thessaloniki, 54124 Thessaloniki, Greece; fountzil@auth.gr; 6Department of Medical Oncology, School of Health Sciences, Faculty of Medicine, Papageorgiou Hospital, Aristotle University of Thessaloniki, 56429 Thessaloniki, Greece; rromanidou@gng.gr

**Keywords:** *CHEK2*, hereditary breast cancer, founder effect, estrogen receptor, genetic testing

## Abstract

**Simple Summary:**

*CHEK2* germline pathogenic variants are identified at a relatively high frequency among hereditary breast cancer cases and are known to be associated with intermediate breast cancer risk i.e., 2–2.5-fold increase, compared to the general population. Histopathological characteristics and clinical outcomes of breast cancer patients who are *CHEK2* carriers have not been thoroughly investigated. We have therefore sought to determine the *CHEK2* variant spectrum and identify variants with possible founder effect, while investigating the clinicopathological features and outcomes of Greek patients who were *CHEK2* carriers. Three variants have been identified as Greek founders. The vast majority of *CHEK2*-associated breast tumors were hormone receptor positive, underlying a possible benefit from chemoprophylaxis with tamoxifen. A trend for longer survival was observed in patients that underwent mastectomy and received hormone-therapy. Nearly half of patients underwent a risk-reducing surgery, which was not mandated according to current guidelines or relevant risks associated with *CHEK2*.

**Abstract:**

*CHEK2* germline pathogenic variants predispose to breast cancer and possibly to other malignancies, with their spectrum and frequency being variable among populations. Τhe majority of *CHEK2*-associated breast tumors are hormone receptor positive; however, relevant clinical outcomes are not well defined. Herein, we illustrate the histopathological characteristics and clinical outcomes of 52 Greek breast cancer patients who are *CHEK2* carriers. Genetic analysis was performed by Sanger/massively parallel sequencing, followed by MLPA. Subsequent haplotype analysis investigated possible founder effects. Blood relatives were offered cascade testing. *CHEK2* variant spectrum was characterized by variability, while influenced by founder effects. The majority of carriers, i.e., 60.8%, were diagnosed with breast cancer before the age of 45. Notably, 91.5% of breast tumors were hormone receptor positive. Hormone therapy and mastectomy at diagnosis seem to have a positive trend on overall survival, after a median follow-up of 9.5 years. Remarkably, 41.9% of patients underwent risk-reducing surgery, one third of which involved salpingo-oophorectomy. Nearly half of families responded to cascade testing. Our data highlight the need for guideline-adherent choices, based on the evidence that *CHEK2* carriers are at moderate risk for breast cancer and no risk for ovarian cancer, while underscore the possible role of chemoprevention with tamoxifen.

## 1. Introduction

The advent of DNA sequencing technologies has enabled the identification of multiple breast cancer-predisposition genes, beyond *BRCA1* and *BRCA2.* Among these, pathogenic variants in *CHEK2*, along with *ATM*, seem to be the most prevalent, known to be associated with intermediate lifetime risks for breast cancer [1,2,3], and are therefore considered moderate penetrance genes. Specifically, germline pathogenic *CHEK2* variants, excluding the low-risk variants, p.(Ile157Thr) and p.(Ser428Phe), known to be associated with much lower risk, confer an approximate lifetime risk for breast cancer diagnosis of about 25–30% in females [4,5]. Identification of cancer-predisposing variants prompts access to customized surveillance protocols; for female *CHEK2* carriers this translates to breast screening by mammography and magnetic resonance imaging [6]. Furthermore, both male and female *CHEK2* carriers seem to have an increased risk for colon, prostate, thyroid and possibly gastric cancer, with risk estimates not being accurate, mainly due to limited data [7,8,9].

The majority of clinical data and *CHEK2*-associated risks derive from the study of the most commonly encountered *CHEK2* variant, i.e., c.1100delC causing frameshift and a premature termination, while risk associations for missense *CHEK2* variants with damaging effect have been proposed to be potentially variable [10]. Missense *CHEK2* variants are individually rare, while their classification and clinical relevance might be unclear. Two, quite recent, large case-control studies have highlighted the clinical implications associated with missense *CHEK2* variants, since they were found to be associated with moderate breast cancer risk, while a strong association of all *CHEK2* pathogenic variants with estrogen receptor positive breast cancer was revealed [4,5]. A number of *CHEK2* founder variants have been already reported in various populations namely, c.444+1G>A, del5395 and deletion of exon 6 [11,12,13], while additional founders are to be characterized. Such studies might provide additional evidence for *CHEK2*-associated tumors.

Herein, we have collected fifty-two breast cancer patients of Greek descent that have been identified as *CHEK2* pathogenic variant carriers through germline genetic testing. We have therefore, described the spectrum of identified *CHEK2* variants, collected all the histopathological characteristics, calculated the clinical outcome, while monitored the choices on uptake of surgical procedures.

## 2. Materials and Methods

### 2.1. Patients Study Group

A total of fifty-two (51 females and one male), apparently unrelated, carriers of germline *CHEK2* pathogenic/likely pathogenic variants, all diagnosed with breast cancer, were included in the study. Carriers of p.(Ile157Thr) were excluded. These carriers have been previously identified through a number of studies, involving genetic testing among Greek patients with breast cancer [11,14,15,16]. All patients fulfilled the National Comprehensive Cancer Network (NCCN) guidelines for genetic testing and were referred to Molecular Diagnostics Laboratory (MDL) of NCSR “Demokritos”, between years 2007–2020, from several collaborating oncology clinics. Detailed pedigrees were obtained through phone interviews.

Clinical and histopathological data were obtained through histology reports. Follow-up data on patient overall survival and updates on personal/family history were collected up until February 2021. Subsequently, cascade testing was offered to blood relatives of *CHEK2* carriers that were informed by the proband of each family, due to the Greek legislation on confidentiality. For all relatives that pursued targeted testing, genetic counseling was carried out over the phone. The study was approved by the Bioethics Committees (Reference number: NCSRD-BC report 14/02/2014), in agreement with the 1975 Helsinki statement, revised in 1983. Signed informed consent was obtained from all participants prior to genetic analysis.

### 2.2. Genetic Analysis of CHEK2 Gene

Genomic DNA and RNA were extracted following standard procedures, as previously described [16]. All DNA samples were either Sanger or massively parallel sequenced, as previously described [16]. Prior to Sanger sequencing, amplification of exons 10–14 of *CHEK2* gene was initially approached by Long-Range PCR to avoid the multiple copies of pseudogenes with high homology located in the region [11].

Variants were described based on NM_007194.4 reference sequence, following Human Genome Variation Society nomenclature. Detection of large genomic rearrangements was performed using the SALSA MLPA kit P190 including *CHEK2* gene, according to manufacturer’s instructions (MRC Holland, Amsterdam, the Netherlands) or by a customized PCR assay, as previously mentioned [11].

### 2.3. Haplotype Analysis

Haplotype analysis was performed using four polymorphic microsatellite markers (D22S1163, D22S689, D22S275, D22S1150), spanning a 1580 kb genomic region on chromosome 22, surrounding the *CHEK2* gene (Appendix A), as already described [11]. More specifically, the analysis included: (i) nine c.499G>A carriers and nine non-carriers, who were blood relatives. (ii) six c.549G>C carriers and three non-carriers, who were blood relatives (iii) five c.592+3A>T carriers and three family relatives that were non-c.592+3A>T carriers. Additionally, one hundred four chromosomes of cancer-free age, matched females were also analyzed to estimate the population allele frequencies of the tested microsatellite markers (Appendix A).

### 2.4. Age Estimation of CHEK2, c.499G>A, c.549G>C and c.592+3A>T Variants

DMLE2.2 software was used to estimate the age that *CHEK2* variants with founder effect initially occurred. This method is based on the observed linkage disequilibrium among a disease variant and linked markers in DNA of allele carriers. The program uses the Markov Chain Monte Carlo algorithm for Bayesian estimation of the variant age [17]. The population growth rate was 0.135 and was based on demographic data, assuming a time interval of 25 years per generation.

### 2.5. Statistical Analysis

Quantitative variables are presented as mean [SD] and categorical variables as percentage. Survival probability was estimated using the Kaplan-Meier method. Survival curves were compared across groups with the log-rank test. Event-free survival (EFS) was defined as the time from diagnosis of breast cancer to diagnosis of a second primary tumor, metastasis, first documented recurrence, death from any cause or last contact, whichever occurred first. Overall survival (OS) was defined as the time from breast cancer diagnosis to the date of death from any cause. Alive patients were censored at the last follow-up date. Survival analysis was performed only for female patients with ER-positive breast cancer diagnoses. *p*-values of <0.05 were considered statistically significant.

## 3. Results

### 3.1. CHEK2 Pathogenic/Likely Pathogenic Variant Spectrum

Among the fifty-two breast cancer patients, twenty distinct pathogenic/likely pathogenic *CHEK2* variants were detected. Of these, seven involved frameshift variants, two were nonsense and six were missense variants, previously classified as damaging, while one involved a splice variant [18]. In addition to that, four *CHEK2* large genomic rearrangements were identified, namely a deletion of exon 1 encompassing the 5′UTR region, a deletion encompassing exons 2 and 3 of the gene [11], the Greek founder deletion of exon 6 [11] and the Czech founder 5.6 kb deletion (also known as del5395) [12]. The schematic representation of all *CHEK2* variants detected among Greek breast cancer patients, is illustrated in Figure 1.

Interestingly, three *CHEK2* variants namely, c.499G>A, c.549G>C and c.592+3A>T were recurrent. Of these, c.499G>A was identified in nine families, while each of the c.549G>C and c.592+3A>T variants, were identified in four families. They were therefore further investigated for their possible founder effect in the Greek population. Moreover, the *CHEK2* c.592+3A>T has been predicted by VarSeak (http://www.varseak.bio/) (accessed 8 May 2020) and NNSplice [19] to affect the 5′ canonical splice donor site, leading to exon skipping, in line with experimental data.

RNA analysis was conducted, where cDNA amplification revealed two fragments of 412 bp and 264 bp (Figure 2a), corresponding to the wild type and mutant allele, respectively, suggesting the variant causes skipping of exon 3, which is located within protein’s FHA domain. This finding was confirmed by Sanger sequencing (Figure 2b,c). A schematic representation of the splicing effect is depicted in Figure 2d.

### 3.2. Clinical and Histopathological Features of CHEK2 Variant Carriers

The mean [SD] age of primary breast cancer diagnosis was 42.84 years [6.79], range: 30–61 years. The vast majority (88.2%; 45/51) of female *CHEK2* carriers were diagnosed with premenopausal breast cancer, with two thirds of them (60.8%; 31/51) being diagnosed at a young age, i.e., <45 years. Moreover, the mean age at primary breast cancer diagnosis for *CHEK2* carriers of truncating variants (excluding carriers of large genomic rearrangements) and *CHEK2* carriers of missense variants was 43.88 years [6.83] and 40.12 years [5.8], respectively. There was no statistical difference between these subgroups.

Overall, a total of 68 breast cancers occurred (including all breast cancer diagnoses and relapses), for which 59 with available histopathology reports. The majority of breast tumors involved ductal carcinomas (84.7%; 50/59), of which 41 were invasive and nine were in situ carcinomas. Moreover, 6.8% and 8.5% were lobular and of mixed histology, respectively. According to available information, 91.5% (54/59) of breast tumors were estrogen receptor (ER) positive and 67.8% (40/59) of tumors were progesterone receptor (PR) positive, whereas HER2 (human epidermal growth factor receptor 2) was overexpressed in 18.6% (11/59) of tumors. Notably, two breast tumors were triple-negative. Nearly half of breast tumors, i.e., 42.4% (25/59), were grade II, while 40.7% were high-grade at diagnosis (24/59). Among carriers of *CHEK2* truncating variants, 50% (12/24) and 46% (11/24) of primary breast tumors were grade III and II, respectively. On the other hand, among carriers of *CHEK2* missense variants, 45% (5/11) and 36% (4/11) of primary breast tumors were grade III and II, respectively. These findings did not reach statistical significance.

In our cohort, ten out of fifty-two (19.2%) *CHEK2* carriers were diagnosed with contralateral breast cancer, of whom one also developed colorectal cancer. Loco-regional relapses were documented in five patients (9.6%), while metastatic disease was diagnosed in 15.4% (8/52) of them within a median follow-up period of 9.5 years (range: 4 months–43 years). None of the patients was diagnosed with ovarian cancer. All clinicopathological information are summarized in Table 1.

### 3.3. Surgical Management

All *CHEK2* carriers, apart from one who was a *de novo* metastatic patient, underwent breast surgery. More specifically, 27 patients underwent lumpectomy, while 24 patients underwent mastectomy, as their primary treatment. According to available information (*n* = 43), a total of eighteen patients (41.9%; 18/43) underwent risk-reducing surgeries. Of these, four and eight underwent contralateral risk-reducing mastectomy and prophylactic salpingo-oophorectomy (RRSO), respectively, while six patients pursued both. Of these, more than one third, i.e., 8/18, pursued further surgical management as a risk-reducing measure, based on their genetic testing result, of which five patients underwent RRSO.

### 3.4. Patient Outcomes

Among the forty-one patients with ER-positive breast tumors, information on hormone therapy was available for thirty-five of them, of which thirty-two (91.4%; 32/35) received hormone therapy. Three patients refused therapy.

The time period for endocrine therapy administration ranged from 6 months to 29 years; the mean time period was 7.3 years. Analyzing a total of forty ER-positive women, both hormone therapy and type of surgery, i.e., mastectomy as initial treatment, seemed to give a positive trend on 10-year event-free survival. Specifically, *CHEK2* carriers who underwent mastectomy and/or received hormone-therapy as initial treatment, had a numerical increase on 10-year event free survival, although, due to small sample size, no statistical significance was reached. A detailed graphical representation is illustrated in Figure 3.

### 3.5. Associations with Family History

Information for family history was available for 49 *CHEK2* carriers. A family history of breast cancer was reported by 63.3% (31/49) of carriers, while 36.7% (18/49) reported no relevant family history.

### 3.6. Cascade Testing

Among the carriers in our cohort, a total of 48 family relatives from 23 families consented on cascade testing, indicating a response rate of 44.2% (23/52). The number of individuals tested per family ranged from one to four. Among the 48 blood relatives, one third and two thirds were men and women, respectively. Of the tested individuals, 19 tested positive for the familial *CHEK2* variant. Overall, ten of the family members tested had been diagnosed with cancer; six with breast cancer, one with breast and endometrial cancer, two with gynecological and one with thyroid cancer. The vast majority of tested individuals, i.e., 89.6% (43/48); were first-degree family relatives, whereas 10.4% (5/48) were second and third degree family members.

### 3.7. Haplotype Analysis of Variants with Possible Founder Effect and Age Estimation

The haplotype analysis for the *CHEK2,* c.499G>A, c.549G>C and c.592+3A>T recurrent variants revealed common shared core disease-associated haplotypes namely, “143-210-168-223”, “143-218-166-221” and “149-206-162-219”, respectively. These occur along a region of 1,580 kb on chromosome 22 between microsatellite markers D22S1163-D22S1150, illustrating the founder effect for the Greek population. (Appendix A). Representative family pedigrees with detailed haplotypes among carriers and non-carriers for these three variants, are illustrated in Figure 4.

The age of the *CHEK2,* c.499G>A, c.549G>C and c.592+3A>T variants was determined approximately at 40 (range: 26–65), 34 (range: 12–60) and 35 (range: 20–62) generations ago, corresponding to 1000 (range: 650–1625), 850 (range: 300–1500) and 875 (range: 500–1550) years, respectively.

## 4. Discussion

In the present study we describe the *CHEK2* variant spectrum, while depicting the histopathological characteristics and clinical outcomes of fifty-two Greek breast cancer patients who were *CHEK2* carriers. In addition to that, we document patient choices following their genetic test results.

Our data demonstrated the identification of twenty distinct deleterious variants among fifty-two *CHEK2* carriers with breast cancer. Deleterious missense variants and large genomic rearrangements, scattered along the sequence of *CHEK2*, accounted for a significant proportion of genetic findings. Four *CHEK2* pathogenic variants namely, c.499G>A, c.549G>C, c.592+3A>T and c.793-1557_847-485del [11] were found to have a founder effect for individuals of Greek descent, sharing the same disease-associated haplotype. Of these variants, the most prevalent were: c.499G>A and c.793-1557_847-485del, which were by far more frequent than the c.1100del variant [14].

*CHEK2* carriers showed similar clinical and histopathological characteristics with those reported in previous studies published [20,21,22]. The vast majority of them was diagnosed with premenopausal breast cancer, in line with two recent studies where the estimated breast cancer risk decreased significantly with increasing age for *CHEK2* carriers [4,5].

We herein provide evidence that both truncating and missense *CHEK2* deleterious variants, predispose more to ER-positive breast cancer, illustrated by the really high representation, i.e., 91.5%, herein. This finding is strongly supported by other recent studies, where a large number of patients have been analyzed and the odds ratio for association with ER-positive breast cancer was calculated ~2.5 [4,5,22]. Triple-negative breast cancers are less likely to be associated with *CHEK2* germline pathogenic variants, as was recently illustrated [4]. In accordance with that, just 3.4% of the breast cancers diagnosed in our series were triple-negative. Our findings regarding hormone receptor expression in this subpopulation of women are in line with previously published data and underscore the possible role of chemoprophylaxis with tamoxifen [13,23,24,25].

We also observed an association with ductal carcinoma diagnosis, which has been also demonstrated by other researchers [20], along with an increased rate, i.e., 19.2%, of contralateral breast cancer diagnoses. This is in concordance with other reports, where rates have been reported to range between 9.4–21% [26,27,28]. Furthermore, in our series, hormone therapy and/or mastectomy at primary diagnoses seem to have a positive effect on survival of *CHEK2* carriers. Of course, these data need to be interpreted with caution due to our small sample size.

The most important finding of our study involves the possible ‘’over-treatment’’ of these patients, in terms of so-called risk-reducing surgical procedures performed. Herein, a significant proportion of our patients, i.e., ~42%, underwent such a procedure, either involving mastectomy or salpingo-oophorectomy, despite the current practice guidelines [6]. Remarkably, 18.6% of patients proceeded with surgery based on their genetic test results, i.e., being a *CHEK2* carrier. Notably, 11.6% of patients proceeded with salpingo-oophorectomy, despite the fact that *CHEK2* pathogenic variants are not known to be associated with increased risk for ovarian cancer [29,30]. In the context of this study, individual interviews with patients that underwent these procedures, have not been scheduled. Therefore, the exact reasons for the uptake of these procedures have not been elucidated. These findings indicate the overuse of prophylactic mastectomy, based on the calculated cumulative breast cancer risks for females with *CHEK2* pathogenic variants, in line with other studies [31,32].

Although it might not be fully informative for genes with intermediate penetrance, cascade testing was offered to blood relatives of all our carriers. Approximately, family relatives from half of the families included in the study proceeded to testing, which is a slightly higher response rate, compared to previous studies [33,34,35]. Females were more likely to undergo testing, compared to males, while rates of testing of individuals beyond first degree kindship was low, presumably due to refusal to disclose results. Cascade testing can be proven a powerful tool for primary cancer prevention, through surveillance of individuals that have increased risk, while providing specific information, which can ultimately result in avoidance of unnecessary interventions. New genetic counseling approaches leading to better family communication and awareness are needed.

Our study has limitations. The most important is our small sample size, which did not permit statistically significant results to emerge, while some clinical information used herein might not be accurate, as it has been recorded directly from patients.

## 5. Conclusions

This study provides an overview of the spectra of *CHEK2* pathogenic variants among Greek breast cancer patients, while sheds some light on the clinicopathological characteristics of these tumors. Most interestingly, our data highlight the need for guideline-adherent choices, based on the evidence that *CHEK2* carriers face a moderately increased risk for breast cancer and no elevated risk for ovarian cancer.

## Figures and Tables

**Figure 1 cancers-13-02106-f001:**
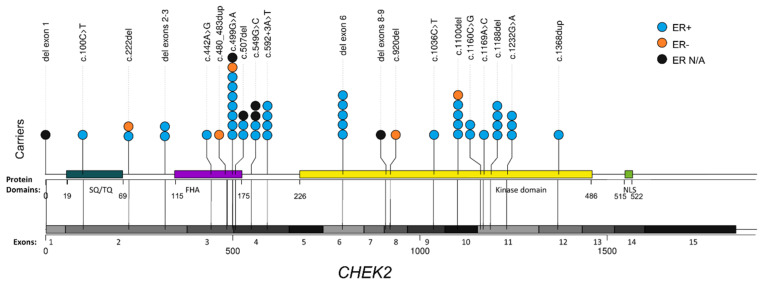
*CHEK2* pathogenic variant spectrum found among Greek breast cancer patients. Abbreviation used: N/A, not available.

**Figure 2 cancers-13-02106-f002:**
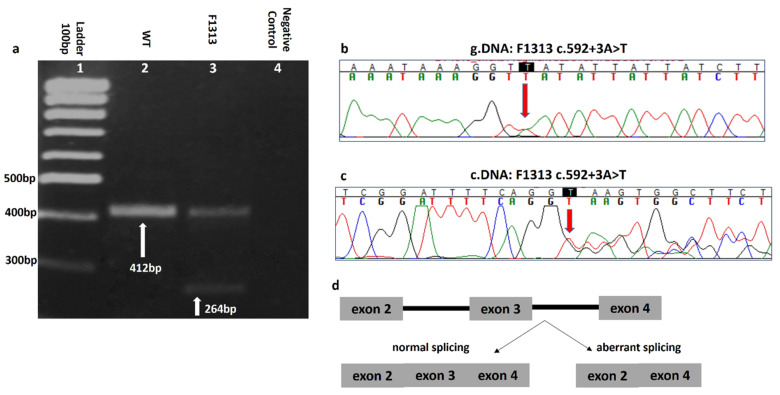
RNA analysis of the *CHEK2,* c.592+3A>T pathogenic variant. (**a**) cDNA products visualized on an agarose gel, where the 412 bp and 264 bp bands correspond to wild-type and mutant allele, respectively, (**b**) Electropherogram of genomic DNA and (**c**) cDNA from the carrier patient, F1313, performed by Sanger sequencing and (**d**) schematic representation of the skipping event.

**Figure 3 cancers-13-02106-f003:**
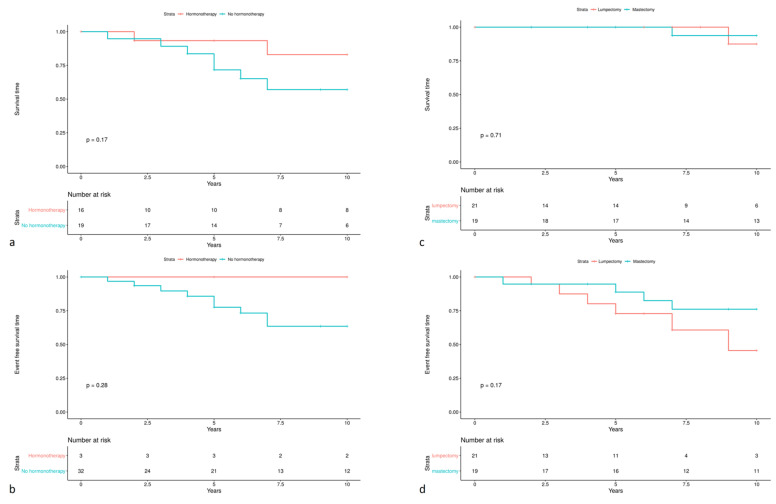
Kaplan-Meier Curves of 10-Year Survival among female patients with ER-positive breast cancer diagnoses, carrying a *CHEK2* pathogenic variant. (**a**) Overall survival defined by administration of hormone-therapy. (**b**) Event-free survival defined by administration of hormone-therapy. (**c**) Overall survival defined by type of breast surgery. (**d**) Event-free survival defined by type of breast surgery.

**Figure 4 cancers-13-02106-f004:**
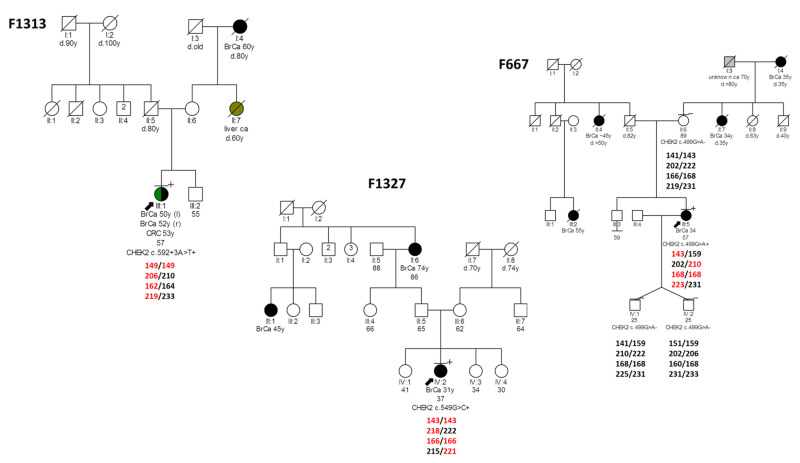
Pedigrees of families, members of which carry the *CHEK2* founder pathogenic variants, where haplotype analysis results are shown. Probands are represented by arrows. Disease-associated alleles are shown in red. Abbreviations used: ca, cancer; BrCa, breast cancer; CRC, colorectal cancer.

**Table 1 cancers-13-02106-t001:** Pathogenic *CHEK2* variants identified in breast cancer cases, along with patients’ clinical and histopathological characteristics.

Patient ID	F/M	Variant (DNA level, NM _007194.4)	BrCa 1st dx (y)	Histology	Grade	ER	PR	HER2	Hormonotherapy (Years)	Surgery	Other ca dx (y)	Prophylactic Mastectomy	RRSO	Distant Met (y)	FH
F876	F	g.(29130716_29137756)_(29137822_?)del * [del exon 1]	47 **	N.A.	N.A.	N.A.	N.A.	N.A.	N.A.	Lumpectomy	relapse 56	N.A.	N.A	N.A	Yes
F993	F	c.100C>T	47	Ductal	2	+	+	-	Yes (14 -in process)	Mastectomy	No	No	No	No	Yes
F1912	F	c.222del	61	Ductal	2	-	-	-	No	Lumpectomy	No	No	No	No	N.A.
F2217	F	c.222del	42	DCIS	3	+	+	-	Yes (5)	Lumpectomy	No	No	No	No	Yes
F1081	F	c.320-4172_593-3895del [del exons 2-3]	36	Ductal	3	+	-	+	Yes (10)	Mastectomy	No	CPM	RRSO	No	No
F3519	F	c.320-4172_593-3895del [del exons 2-3]	41	Mixed (ductal/lobular)	2	+	+	N.A.	Yes (2 -in process)	Lumpectomy	No	No	No	No	No
F5055	F	c.442A>G	39	Ductal	3	+	+	-	Yes (3 -in process)	Mastectomy	No	No	No	No	No
F508	F	c.480_483dup	39	Mixed (ductal/lobular)	2	-	-	N.A.	No	Μastectomy	BrCa 58	CPM	RRSO	No	Yes
F667	F	c.499G>A	34	Ductal	1	+	+	N.A.	No	Lumpectomy	No	No	No	No	Yes
F1326	F	c.499G>A	38	N.A.	N.A.	N.A.	N.A.	N.A.	N.A.	Lumpectomy	N.A.	N.A.	N.A	N.A	Yes
F2050	F	c.499G>A	46	Ductal	1	+	+	-	Yes (10)	Mastectomy	No	No	RRSO	No	Yes
F2276	M	c.499G>A	41	DCIS	N.A.	+	+	-	N.A.	Lumpectomy	relapse 45	N.A.		N.A.	N.A.
F2469	F	c.499G>A	35	Ductal	3	+	+	-	No	Lumpectomy	No	PBM	No	No	No
F2605	F	c.499G>A	39	Ductal	3	-	-	-	No	Mastectomy	No	CPM	No	No	No
F2774	F	c.499G>A	38	Ductal	2	+	-	-	Yes (5)	Mastectomy	BrCa 44	No	RRSO	No	No
F3445	F	c.499G>A	50	DCIS	3	+	+	+	Yes (5)	Mastectomy	No	CPM	RRSO	No	Yes
F3754	F	c.499G>A	40	Ductal	2	+	-	-	Yes (1 -in process)	Lumpectomy	No	No	No	No	Yes
F322	F	c.507del	51	Ν.A.	Ν.A.	N.A.	N.A.	N.A.	Ν.A.	Lumpectomy	BrCa 67	Ν.A.	Ν.A.	Ν.A.	Yes
F2305	F	c.507del	41	DCIS	3	+	+	-	Yes (4)	Lumpectomy	relapse 45	PBM	No	No	Yes
F3127	F	c.507del	44	Ductal	3	+	-	-	Yes (5)	Lumpectomy	BrCa 49	No	No	No	No
F459	F	c.549G>C	46	N.A.	N.A.	N.A.	N.A.	N.A.	N.A.	Lumpectomy	N.A.	N.A.	N.A.	N.A.	Yes
F498	F	c.549G>C	41	N.A.	N.A.	N.A.	N.A.	N.A.	N.A.	Bil-Mastectomy	No	PBM	RRSO	Yes (69)	No
F1018	F	c.549G>C	36	Ductal	N.A.	+	+	-	Yes (8)	Lumpectomy	relapse 39	No	No	No	Yes
F1327	F	c.549G>C	31	DCIS	2	+	+	-	Yes (10)	Mastectomy	No	No	No	No	Yes
F1313	F	c.592+3A>T	50	Lobular	1	+	+	-	Yes (10)	Lumpectomy	BrCa 52 CRC 53	No	No	No	Yes
F2482	F	c.592+3A>T	33	Ductal	3	+	+	-	Yes (10)	Mastectomy	No	No	No	No	Yes
F3247	F	c.592+3A>T	36	DCIS	2	+	+	-	Yes (5)	Lumpectomy	No	No	No	No	No
F3710	F	c.592+3A>T	50	Ductal	3	+	+	N.A.	Yes (6 months -in process)	Lumpectomy	No	No	No	No	Yes
F1070	F	c.793-1557_847-485del [del exon 6]	57	Lobular	3	+	+	+	Yes (5,5)	Mastectomy	No	No	RRSO	No	No
F1136	F	c.793-1557_847-485del [del exon 6]	42	Ductal	2	+	+	+	N.A.	Lumpectomy	No	N.A.	RRSO	Yes (48)	Yes
F1933	F	c.793-1557_847-485del [del exon 6]	55	Ductal	2	+	-	+	No	Mastectomy	No	No	No	No	Yes
F1937	F	c.793-1557_847-485del [del exon 6]	38	Ductal	2	+	+	+	Yes (3)	Mastectomy	BrCa 50	No	No	No	Yes
F1938	F	c.793-1557_847-485del [del exon 6]	48	Lobular	2	+	+	-	Yes (5)	Mastectomy	No	No	No	No	Yes
F2034	F	c.909-2028_1096-698del [del exons 8-9]	42	N.A.	N.A.	N.A.	N.A.	N.A.	Yes (N.A.)	Mastectomy	No	No	RRSO	No	Yes
F5540s	F	c.920del	44	Ductal	3	-	-	+	No	No	No	No	No	Yes (44)	No
F1741	F	c.1036C>T	40	Lobular	3	+	+	-	Yes (5)	Mastectomy	No	No	No	Yes (45)	Yes
F1556	F	c.1100del	41	Mixed (ductal/lobular)	2	+	+	-	Yes (8)	Mastectomy	BrCa 42	No	No	No	Yes
F1629	F	c.1100del	45	Mixed (ductal/lobular)	3	+	-	-	Yes (29 -in process)	Mastectomy	relapse 50 & 63	No	RRSO	No	No
F1630	F	c.1100del	49	Ductal	2	+	+	+	N.A.	Lumpectomy	N.A.	N.A.	N.A.	N.A.	No
F1631	F	c.1100del	47	Ductal	3	-	-	+	No	Lumpectomy	N.A.	N.A.	N.A.	N.A.	No
F3412	F	c.1100del	37	Ductal	3	+	+	-	Yes (5)	Mastectomy	BrCa 50	No	No	No	No
F619	F	c.1160C>G	43	Ductal	N.A.	+	-	-	N.A.	Lumpectomy	N.A.	N.A.	N.A.	Yes (43)	Yes
F1960	F	c.1160C>G	53	Ductal	N.A.	+	+	N.A.	Yes (6 months)	Mastectomy	No	No	No	No	Yes
F3477	F	c.1169A>C	33	Ductal	2	+	-	-	Yes (10)	Μastectomy	Νο	CPM	No	No	Yes
F2839	F	c.1188del	30	Ductal	3	+	+	-	N.A.	Lumpectomy	N.A.	N.A.	N.A.	N.A.	N.A.
F2891	F	c.1188del	47	Ductal	2	+	N.A.	N.A.	Yes (7)	Lumpectomy	BrCa 63	No	RRSO	No	No
F2983	F	c.1188del	47	Ductal	2	+	+	+	Yes (5)	Mastectomy	No	CPM	RRSO	Yes (51, 58)	Yes
F3696	F	c.1188del	35	Ductal	2	+	+	-	Yes (10)	Mastectomy	No	No	No	Yes (49)	No
F2261	F	c.1232G>A	53	Ductal	3	+	+	-	N.A.	Lumpectomy	No	CPM	RRSO	No	Yes
F2790	F	c.1232G>A	38	Ductal	2	+	N.A.	N.A.	Yes (14)	Lumpectomy	BrCa 45	No	RRSO	Yes (52)	No
F3158	F	c.1232G>A	42	Ductal	3	+	+	-	Yes (10)	Lumpectomy	No	No	No	No	Yes
F802	F	c.1368dup	48	Ductal	2	+	+	-	Yes (10)	Lumpectomy	No	No	No	No	Yes

Abbreviations used: BrCa, breast cancer; ca, cancer; CPM, contralateral prophylactic mastectomy; CRC, colorectal cancer; DCIS, ductal carcinoma in situ; dx, diagnosis; ER, estrogen receptor; F, female FH; family history; HER2, human epidermal growth factor receptor 2; M, male; N.A., not available; PBM, prophylactic bilateral mastectomy; PR, progesterone receptor; RRSO, risk-reducing salpingo-oophorectomy; y, years; +, positive, -, negative amily history yes: at least one family relative with diagnosis of breast cancer. * According to HGVS recommendations, deletions extending beyond the transcribed region can only be described using genomic coordinates. ** diagnosis of bilateral breast cancer.

## Data Availability

All data that support the findings of our study are available upon reasonable request.

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
