# Peer review of "CHEK2 Pathogenic Variants in Greek Breast Cancer Patients: Evidence for Strong Associations with Estrogen Receptor Positivity, Overuse of Risk-Reducing Procedures and Population Founder Effects"

_cancers, 2021, doi:10.3390/cancers13092106_

Round 1

Reviewer 1 Report

Comments about the research work entitled CHEK2 pathogenic variants in Greek breast cancer patients: evidence for strong associations with estrogen receptor positivity, overuse of risk-reducing procedures and population founder effects

I consider this research a particularly important work to help support treatment and procedural decisions in order to establish a relationship between histopathology and clinical outcomes of patients with any variant associated with pathogenicity.

The researchers of this work have managed to determine a spectrum of variants, some with a possible founder effect, and even more interesting, clinicopathological characteristics of Greek patients who carry these variants. Three of these variants have been identified as Greek founders. The vast majority of these CHEK2 variants were associated with hormone receptor-positive tumors. Of 52 breast cancer patients, they found relevant results such as CHEK2 in germline predisposes to breast cancer among other neoplasms with a spectrum and frequency variable between populations. Most of the 60% carriers were diagnosed with cancer before age 45. About 90% were hormone receptor-positive.

It is a rigorously designed work, clearly explained and I agree with the authors, after reading their work, that the data provided from this study supports compliance with treatment guidelines, based on the evidence that carriers of CHEK2 are at risk of moderate breast cancer

The introduction has a strong justification for the work and the importance of studying the risk associated with missense CHEK2 variants, especially when other studies have shown similar results. Studies on founder mutations in this gene as in others of importance in the participation of neoplasms, are necessary for the different populations considering that the treatments tend to be more and more individualized.

The methods used are rigorous and the achievement of at least 52 patients with breast cancer who meet the criteria of being unrelated and, germline carriers of pathogenic CHEK2 variants.

Comments with minor adjustments

3.1 The description of the variants found is clear and Figure 1 is very illustrative and summarizes what was found in the text. In this regard, I have only one suggestion and that is that within the text of figure 1, expand the meaning of the black ovals that in the figure are only shown as ER N/A, do the same as appears later in the text of the table 1.

Another suggestion for this section is to check line 173 "identified in nine, four and four unrelated families, respectively" which generates confusion.

Figure 2 is especially useful and supports what corresponds to the variant that affects the domain of the protein.

3.2 Just to specify the information on line 197, make it clear that the 52 patients are referred to.

From the discussion, the contribution of these results to support the treatment and procedures to be carried out among patients with these pathogenic variants is noteworthy.

Author Response

Authors responses to reviewers’ comments

- manuscript cancers-1186018 

Reviewer #1: Comments to the Author

I consider this research a particularly important work to help support treatment and procedural decisions in order to establish a relationship between histopathology and clinical outcomes of patients with any variant associated with pathogenicity.

The researchers of this work have managed to determine a spectrum of variants, some with a possible founder effect, and even more interesting, clinicopathological characteristics of Greek patients who carry these variants. Three of these variants have been identified as Greek founders. The vast majority of these CHEK2 variants were associated with hormone receptor-positive tumors. Of 52 breast cancer patients, they found relevant results such as CHEK2 in germline predisposes to breast cancer among other neoplasms with a spectrum and frequency variable between populations. Most of the 60% carriers were diagnosed with cancer before age 45. About 90% were hormone receptor-positive.

It is a rigorously designed work, clearly explained and I agree with the authors, after reading their work, that the data provided from this study supports compliance with treatment guidelines, based on the evidence that carriers of CHEK2 are at risk of moderate breast cancer

The introduction has a strong justification for the work and the importance of studying the risk associated with missense CHEK2 variants, especially when other studies have shown similar results. Studies on founder mutations in this gene as in others of importance in the participation of neoplasms, are necessary for the different populations considering that the treatments tend to be more and more individualized.

The methods used are rigorous and the achievement of at least 52 patients with breast cancer who meet the criteria of being unrelated and, germline carriers of pathogenic CHEK2 variants. Figure 2 is especially useful and supports what corresponds to the variant that affects the domain of the protein. From the discussion, the contribution of these results to support the treatment and procedures to be carried out among patients with these pathogenic variants is noteworthy.

We would really like to thank reviewer #1 for the positive comments on our work.

-The description of the variants found is clear and Figure 1 is very illustrative and summarizes what was found in the text.

 In this regard, I have only one suggestion and that is that within the text of figure 1, expand the meaning of the black ovals that in the figure are only shown as ER N/A, do the same as appears later in the text of the table 1.

Following the reviewer’s recommendations, the meaning of the abbreviation N/A (Abbreviation used: N/A, not available) was added within the legend text of figure 1.    

 -Another suggestion for this section is to check line 173 "identified in nine, four and four unrelated families, respectively" which generates confusion.

This sentence was rephrased in order to be clarified. More specifically, the sentence is now written as: ‘’Interestingly, three CHEK2 variants namely, c.499G>A, c.549G>C and c.592+3A>T were recurrent. Of these, c.499G>A was identified in nine families, while each of the c.549G>C and c.592+3A>T variants, were identified in four families’’. 

-Just to specify the information on line 197, make it clear that the 52 patients are referred to.

This sentence was rephrased in order to be clarified. The sentence is now written as: “In our cohort, ten out of fifty-two (19.2%) CHEK2 carriers”.

Reviewer 2 Report

The manuscript describes  CHEK2 germline pathogenic variants that has been found at relatively high frequency among hereditary breast cancer. Three variants have been found as Greek founders. Moreover, authors have shown a possible benefit from mastectomy and hormone therapy for survival of patients who were CHEK2 carriers. The manuscript includes Table 1 , which contains characteristic of each patients according to pathogenic CHEK2 variants and clinical and histopathological characteristic. I suppose, if it would possible, to summarize the data in Table 1, i.g. the type of genetic variant of CHEK2 and the type of breast cancer, grade or age etc. I  realizes that the group of patients was small and such summing up may be difficult.

Author Response

Reviewer #2: Comments to the Author

The manuscript describes CHEK2 germline pathogenic variants that has been found at relatively high frequency among hereditary breast cancer. Three variants have been found as Greek founders. Moreover, authors have shown a possible benefit from mastectomy and hormone therapy for survival of patients who were CHEK2 carriers.

We would like to thank the reviewer for the kind comments on our work.

-The manuscript includes Table 1, which contains characteristic of each patients according to pathogenic CHEK2 variants and clinical and histopathological characteristic. I suppose, if it would possible, to summarize the data in Table 1, e.g. the type of genetic variant of CHEK2 and the type of breast cancer, grade or age etc. I realize that the group of patients was small and such summing up may be difficult.

We would like to thank you for that suggestion. We have made an effort to summarize the data on Table 1 and we have added the results in the respective paragraphs. More specifically, we added the following:

  • Section 3.2: “Moreover, the mean age at primary breast cancer diagnosis for CHEK2 carriers of truncating variants (excluding carriers of large genomic rearrangements) and CHEK2 carriers of missense variants was 43.88 years [6.83] and 40.12 years [5.8], respectively. There was no statistical difference between these subgroups”.

“Among carriers of CHEK2 truncating variants, 50% (12/24) and 46% (11/24) of primary breast tumors were grade III and II, respectively. On the other hand, among carriers of CHEK2 missense variants, 45% (5/11) and 36% (4/11) of primary breast tumors were grade III and II, respectively.  These findings did not reach statistical significance”.